# In Vitro Antithrombotic, Antitumor and Antiangiogenic Activities of Green Tea Polyphenols and Its Main Constituent Epigallocatechin-3-gallate

Jefferson Romáryo Duarte da Luz [1,2], Jorge A. López [1,2], Macelia Pinheiro Ferreira [2], Rubiamara Mauricio de Sousa [2,3], Saulo Victor e Silva [2,4], Maria das Graças Almeida [2,3,4] and Gabriel Araujo-Silva [1,*]

[1] Organic Chemistry and Biochemistry Laboratory, State University of Amapá (UEAP), Av. Presidente Vargas, s/n-Centro, Macapá 68900-070, AP, Brazil

[2] Multidisciplinary Research Laboratory, Department of Clinical and Toxicological Analyses DACT, Health Sciences Center, Federal University of Rio Grande do Norte, R. Gen. Gustavo Cordeiro de Farias, s/n-Petrópolis, Natal 59012-570, RN, Brazil

[3] Postgraduate Program in Health Sciences, Health Sciences Center, Federal University of Rio Grande do Norte, R. Gen. Gustavo Cordeiro de Farias, s/n-Petrópolis, Natal 59012-570, RN, Brazil

[4] Postgraduate Program in Pharmaceutical Sciences, Health Sciences Center, Federal University of Rio Grande do Norte, R. Gen. Gustavo Cordeiro de Farias, s/n-Petrópolis, Natal 59012-570, RN, Brazil

[*] Correspondence: gabriel.silva@ueap.edu.br

**Abstract:** The balance between embolic risk and bleeding represents a clinical challenge in cancer patient treatment, encouraging studies on adjuvant oncologic treatments. Thereby, this study evaluated the in vitro effect of green tea extract (GTE) and epigallocatechin-3-gallate (EGCG) on hemostasis modulation and the antineoplastic effect on melanoma cells (B16-F10) by applying platelet aggregation, angiogenesis and viability cell assays. The results displayed a significant platelet antiaggregant effect, corresponding to 50 and 80% for the extract and EGCG, respectively, compared to the negative control. Furthermore, both GTE and EGCG exhibited antitumor effects by reducing melanoma cell growth by 25 and 50%, respectively, verified by cellular apoptosis. Regarding angiogenesis, these substances inhibited blood vessel formation, reaching about 25% and 99% for GTE and EGCG at 100 μg/mL, respectively. Moreover, TNF-α cell stimulation evidenced VEGF and IL-8 secretion inhibition at 55 and 20% with GTE, while EGCG promoted an inhibition around 78% for both VEGF and IL-8. The results indicate the promising performance of GTE and EGCG as an option for treating cancer and its side effects. Nonetheless, further studies are required to elucidate their action mechanism on clotting, cell death and angiogenesis.

**Keywords:** melanoma; cancer; anticoagulant; apoptosis; toxicity; green tea

## 1. Introduction

Overall, cancer is one of the most challenging contemporary diseases in clinical and research aspects due to worldwide morbidity and mortality [1]. Global estimates of the cancer burden indicate a growing incidence, representing one of the main causes of mortality worldwide, causing about 10 million deaths in 2020 [2,3]. Specifically, concerning skin cancer, a forecast report indicates around 27 million new cases by 2030, including melanoma and nonmelanoma skin cancer. Melanoma is responsible for 5% of skin cancer cases, corresponding to 46% of deaths [4–6]. This situation has promoted extensive studies in the search for therapeutic approaches for cancer prevention and treatment [7,8].

Regarding the skin cancer burden, this mutagenic process from healthy melanocytes into cancer cells is attributed to a precancerous lesion that promotes continual unregulated cell proliferation toward malignant tumor development with a high risk of metastatic spread [9]. These changes result from the interaction between genetic factors and external

agents, such as ultraviolet radiation (UV), one of the main factors for its development. Thereby, the MC1R gene polymorphism, a signaling receptor for melanin production, is responsible for encoding the pigment eumelanin or pheomelanin prevalence. Pheomelanin is the largest producer of reactive oxygen species (ROS), whose exposure to UV rays promotes intense DNA damage with consequent cellular mutagenicity [10].

Despite the increased incidence of malignant melanoma, understanding the main oncogenes and signaling pathways involved in its pathogenesis and progression has been a cornerstone to improve treatment outcomes [11]. Concerning treatments, surgical resection was, for a long time, the only therapeutic choice until the later introduction of chemotherapy, radiotherapy and targeted therapies aimed at arresting the disordered proliferation of cancerous melanocytes to prevent the progression of the disease [12]. Nonetheless, these treatments cause several side effects, such as a decrease in immune system function and resistance to therapy, impacting bone marrow cell production and patient clinical management [13,14].

Likewise, thrombosis risk is another side effect of conventional cancer treatments, which is the leading cause of death in outpatients undergoing systemic cancer therapy [15]. This is a consequence of using venous catheters, antiangiogenic agents, hormone therapy, radiotherapy, immunomodulatory drugs and prolonged bed rest during the treatment regimen, with a resulting propensity for hypercoagulability, endothelial injury and venous stasis [16,17]. Considering the high susceptibility to thromboembolism, several approaches have been analyzed in the search for new molecules for oncological therapy [7]. Thereby, therapies involving medicinal plant use have been reported, evaluating them individually or in combination with conventional drugs to enhance efficacy and reduce side effects for the patient under treatment [8,18,19].

Hence, the chemical diversity of natural products of plant origin represents a rich reservoir of bioactive compounds, which can be used as extracts or isolated molecules to assess their potential pharmaceutical applications as anticoagulants and anticancer agents [20–23]. Nowadays, a panel of marketed anticancer drugs is related to natural products, which stimulates their prospection in the probing of anticancer compounds [24]. Furthermore, studies have described the anticoagulant effect of plant extracts due to the presence of several secondary metabolites, such as phenolics, terpenoids, alkaloids, saponins, glucosinolates, anthocyanins and catechins [25,26].

Thence, scientific efforts have reported the clinical use of natural products as a therapeutic approach in cancer treatment to minimize collateral effects concerning normal cells. Therefore, studies have disclosed the antineoplastic and antithrombotic effects of several polyphenols (e.g., flavonoids and catechin). Overall, polyphenols are the subject of studies to develop drugs since they are involved in the direct or indirect inhibition or activation of relevant cellular and molecular targets [27,28].

At this point, green tea is a natural product obtained from the processing of *Camellia sinensis* leaf and is popularly consumed worldwide, which represents an important source of small bioactive molecules. Therefore, studies have pointed out its beneficial effects on human health to treat several diseases (e.g., cancer, diabetes and cardiovascular problems) [29,30]. Green tea phytocomposition plays an essential role in these benefits owing to secondary metabolites, such as polyphenols, lignin, caffeine, organic acids and chlorophyll. Polyphenols are the main functional chemical constituents involved in these pharmacological effects, highlighting the catechin-rich content [29,31]. The profile of the chemical composition of green tea has been elucidated by applying precise analytical methods to recover and detect its components as a strategy to establish product quality standards based on its bioactive constituent importance for health [32,33].

Catechins are the main polyphenolic compound group in green tea, including epigallocatechin-3-gallate (EGCG), epigallocatechin, epicatechin-3-gallate and epicatechin, as well as gallocatechins and gallocatechin gallate. EGCG is the main active compound and the most analyzed one in green tea extract, corresponding to approximately 59% of the total catechin content [29]. Studies suggest that EGCG is an active compound in green tea with a proven

role in curing and preventing human disease by acting as antioxidant, antiangiogenic and antitumor agents and as a modulator of tumor cell response to chemotherapy [26,34]. Moreover, studies have indicated the anticoagulant effect of GTE due to its high EGCG content [35]. Despite the antineoplastic and anticoagulant roles of these compounds, studies regarding the combined assessment of these effects are even scarce.

Overall, studies on the potential of natural plant products are relevant as an approach to cancer treatment, aiming at a possible pharmacological application. Hence, the present in vitro study was designed to evaluate GTE and EGCG effects on both hemostasis and tumor progression in the melanoma cell line (B16-F10) by examining the apoptotic effect of these compounds.

## 2. Materials and Methods

### 2.1. Materials and Reagents

Green tea (*Camellia sinensis*) extract (GTE) (NIST® SRM® 3255), epigallocatechin-3-gallate (EGCG) (CAS No.: 989-51-5), bovine serum albumin (BSA), Dulbecco's Modified Eagle Medium (DMEM), fetal bovine serum (FBS), cisplatin and melanin were purchased from Sigma-Aldrich (São Paulo, Brazil). Low molecular weight heparin (enoxaparin) (MW ~4.5 kDa) was obtained from Sanofi-Aventis Farmacêutica (São Paulo, SP, Brazil). Alamar Blue® and MTT (3-4,5-dimethyl-thiazol-2-yl-2,5-diphenyltetrazolium bromide) were obtained from Invitrogen (Carlsbad, CA, USA), while penicillin–streptomycin (10,000 U/mL) and Ham's F-12 Nutrient Mix were obtained from ThermoFisher Scientific (Waltham, MA, USA). Murine melanoma cell line B16F10 (ATCC CRL-6475) and rabbit aortic endothelial cells (RAEC) were purchased from the American Type Culture Collection (Rockville, MD, USA). All other chemicals and reagents were analytical grade.

### 2.2. Aggregation Platelet Test

This assay was performed by the turbidimetric method using an aggregometer (Chrono-Log, Havertown, PA, EUA). The platelet-rich plasma (PRP) fraction dilution was performed with Tyrode's solution containing 0.3% bovine serum albumin (BSA), adjusting it to a final volume of 400 μL ($2 \times 10^8$ platelets/mL). Different aggregation agents were used to stimulate the PRP aggregation with the final concentrations of 2 μg/mL collagen, 10 μM adenosine diphosphate (ADP), 100 μM arachidonic acid (AA) and 0.2 U/mL thrombin. After 5 min of stimulation, platelet aggregation was recorded, expressing the values as percentage changes in light transmission compared to platelet-free plasma (PFP) as a control. For the in vitro assay, PRP was individually pre-incubated at 37 °C for 5 min with GTE (25, 50 and 100 μg/mL) and with EGCG (25, 50 and 100 μg/mL) or aspirin before stimulation with the aggregation-inducing agents [36].

### 2.3. Activated Partial Thromboplastin Time (aPTT) Assay

The aPTT assay was performed using aPTT kit (WAMA Diagnóstica, São Carlos, SP, Brazil), conforming to the manufacturer's instructions. GTE and EGCG were individually dissolved in a physiological saline solution at 25, 50 and 100 μg/mL, inoculating 10 μL into 90 μL of plasma. After incubation (37 °C/3 min), 100 μL of bovine cephalin was added, followed by incubation at 37 °C for 3 min. Then, 100 μL of pre-warmed 0.25 M CaCl$_2$ solution was added to the mixture before determining the clotting time in triplicate using a Clot Timer Coagulometer (Drake Electronica Commerce Ltd.a., São Paulo, Brazil) [37].

### 2.4. Prothrombin Time (PT) Assay

Clotting time was measured using the TP kit, following the manufacturer's instructions (CLOT Bios Diagnostica, São Paulo, SP, Brazil). GTE, EGCG and heparin were individually dissolved in a physiological saline solution at concentrations of 25, 50 and 100 μg/mL. Then, 10 μL of each dilution was mixed with 90 μL of plasma, and the reaction system was incubated at 37 °C for 3 min before adding 200 μL of Soluplastin reagent. PT was analyzed

in triplicate on an automated Clot Timer Coagulometer (Drake Electronica Commerce Ltd., São Paulo, Brazil) [37].

### 2.5. Cell Culture

Murine melanoma cell line B16F10 was cultured in Dulbecco's Modified Eagle Medium (DMEM), supplemented with 10% fetal bovine serum (FBS), L-glutamine and streptomycin (5000 mg/mL)/penicillin (5000 IU). The RAEC cultivation was carried out in F-12 medium supplemented with 10% FBS, 20 mM sodium bicarbonate and antibiotics. Both cells were incubated at 37 °C in a humidified atmosphere with 5% $CO_2$.

### 2.6. Cell Viability by MTT and Alamar Blue® Tests

Melanoma cell line B16F10 ($1 \times 10^4$ cells/well) was seeded in 96-well microplates and incubated in a humidified atmosphere with 5% $CO_2$ (37 °C/24 h) to promote adhesion. Afterward, cells were individually challenged in triplicate at 25, 50 and 100 μg/mL of GTE and EGCG dissolved in DMEM medium, and the plates were incubated (37 °C/4 h). After this incubation, 10 μL of MTT (5 mg/mL) was added per well-containing cells incubated again under the same conditions. After removing the culture medium, 100 μL of DMSO was added to each well before determining cell viability at 570 nm. The GTE and EGCG cytotoxic effects on cell viability were also evaluated in triplicate using Alamar Blue® assay under the same culture and challenge conditions described above. After exposure to the substances, 10 μL of Alamar Blue® (400 μg/L) dissolved in PBS was added to each well and plates were incubated (37 °C/24 h) before monitoring the reduced Alamar Blue® to 570 and 600 nm. Absorbances were measured in a microplate ELISA reader (Epoch-Biotek, Winooski, VT, USA), using cells grown in DMEM as a negative control [38].

### 2.7. Apoptosis (Annexin-V/Propidium Iodide) Analysis Assay

B16F10 melanoma cells ($2 \times 10^5$ cells/well) were grown in 6-well plates and individually challenged with 100 μg/mL of GTE and EGCG dissolved in DMEM medium by 24 h. Cells grown in DMEM medium were used as a negative control. Then, cells were trypsinized, centrifuged ($800 \times g$/4 °C/5 min) and washed twice with PBS before labeling them with FITC-conjugated Annexin-V and Propidium Iodide using the Apoptosis Detection Kit from Annexin-V FITC, conforming to the manufacturer's specifications. Labeled cells were quantified using a flow cytometer (FACSCanto II, BD Biosciences, Eugene, OR, USA) with FACSDiva software, version 6.1.2 (Becton Dickson, Franklin Lakes, NJ, USA). Data were analyzed using FlowJo software (FlowJo, Ashland, OR, USA) [39]. Cisplatin was used as an antineoplastic drug control under the same experimental conditions.

### 2.8. Apoptosis Indicators Evaluation by Incubation with DAPI

The melanoma cell line B16F10 ($35.55 \times 10^4$ cells/well) was cultured on circular coverslips (13 mm) in a 24-well plate at 37 °C for 45 min. Then, DMEM supplemented with 10% FBS was added to a final volume of 1 mL, and the system was incubated in a humidified atmosphere of 5% $CO_2$ (37 °C/24 h). Thereafter, the medium was removed and cells were incubated for 24 h with serum-free medium. After individual treatment with medium supplemented with GTE and EGCG at 100 μg/mL, cells were washed with cold PBS, fixed with 4% paraformaldehyde for 20 min and permeabilized in 0.1% Triton X-100 (~20 min). Next, cells were washed again with PBS and incubated with 4′,6-diamidino-2-phenylindole (DAPI) at 1 mg/mL for 30 min at room temperature in the dark. Cells were analyzed by fluorescence microscopy with a fluorescence filter at 330–380 nm (OLYMPUS BX41 fluorescence microscope) [40]. GTE and EGCG were dissolved in DMEM medium.

### 2.9. Melanin Determination Assay

Melanin content in melanoma cells was determined by the method described by Palhares et al. [39], with some modifications. Briefly, B16-F-10 cells ($5 \times 10^4$ cells/well) were seeded in 6-well plates and individually challenged with EGCG and GTE (25, 50 and

100 µg/mL) dissolved in DMEM medium, using cells grown in DMEM as a negative control. After 48 h of treatment, cells were trypsinized and centrifuged ($1600 \times g/4\ °C/5$ min). The cell pellet was washed twice with PBS and centrifuged again ($1600 \times g/4\ °C/5$ min) before lysing cells with 1M NaOH for 30 min to release melanin. Melanin content was determined at 475 nm on an Epoch ELISA reader microplate (Epoch-Biotek, Winooski, VT, USA) against a synthetic melanin standard curve. Results were expressed as the melanin percentage produced compared to the melanin content of cells cultured with DMEM (negative control).

### 2.10. Matrigel Endothelial Cell Tube Formation Assay

The EGCG and GTE potential antiangiogenic effect was evaluated in 24-well plates coated with 200 µL of a solubilized basement membrane preparation (Matrigel) and then incubated at 37 °C for 16 h for gelification, as described by Palhares et al. [39]. Next, RAEC cells ($1.0 \times 10^5$ cells/well) were seeded with F-12 medium containing 10% FBS until cell adhesion before adding F-12 medium with 10% FBS and heparin (100 µg/mL) individually supplemented with EGCG and GTE at different concentrations (25, 50, 100 µg/mL) dissolved in F-12 medium. Cells grown in the culture medium alone were used as a control. Triplicate cultures for each treatment were incubated at 37 °C for 24 h in a humidified atmosphere with 2.5% $CO_2$. The formation of the tubular structure was analyzed by an inverted microscope at $100\times$ magnification. Four images were recorded randomly in different areas and analyzed by two different observers. Data obtained from capillary structures on Matrigel were determined and evaluated by Image J software (NIH, Bethesda, MD, USA).

### 2.11. Inhibition of Vascular Endothelial Growth Fator (VEGF) and IL-8 Secretion

Cell culture supernatants in triplicate were collected 48 h after stimulation with EGCG and GTE, centrifuged ($1000 \times g/10$ min) and stored at $-70$ °C. VEGF and IL-8 levels were determined in 96-well microtiter plates using enzyme-linked immunosorbent assays (ELISA; R&D Systems, Abingdon, UK) according to the manufacturer's instructions. The calculation of VEGF growth factor and IL-8 chemokine levels was performed from standard curves using recombinant human VEGF (15.6 to 1000 pg/mL) and recombinant human IL-8 (31.2 to 2000 pg/mL), respectively. Data were obtained from determinations performed in duplicate and expressed in pg/mL per $10^6$ cells [41].

### 2.12. Statistical Analysis

The Shapiro–Wilk test was performed to verify the normal distribution of variables. Then, data expressed as mean $\pm$ SD were analyzed by one-way ANOVA and Tukey's post hoc test. Statistical significance was set at $p < 0.05$. $IC_{50}$ value was determined from the equation formed from the linear regression curve between probit percent inhibition and log concentration. The linear equation obtained is used to find the $IC_{50}$ value. All analyses were performed using GraphPad Prism version 5.0 for Windows (GraphPad Software, San Diego, CA, USA).

## 3. Results

### 3.1. Primary Hemostasis (Platelet Aggregation)

Figure 1 depicts the platelet aggregation inhibitory effect of GTE and EGCG using a platelet-rich plasma (PRP) fraction to assess one of the initial hemostatic mechanisms. As shown in Figure 1A, EGCG displayed a significant antiaggregating effect ($p < 0.05$) compared to GTE, mainly at 100 µg/mL. Nevertheless, the platelet aggregation inhibitory action of aspirin (control) was significantly superior for all evaluated EGCG and GTE concentrations, showing $IC_{50} = 24.124 \pm 0.8$ and $22.254 \pm 0.7$ µg/mL to GTE and EGCG, respectively.

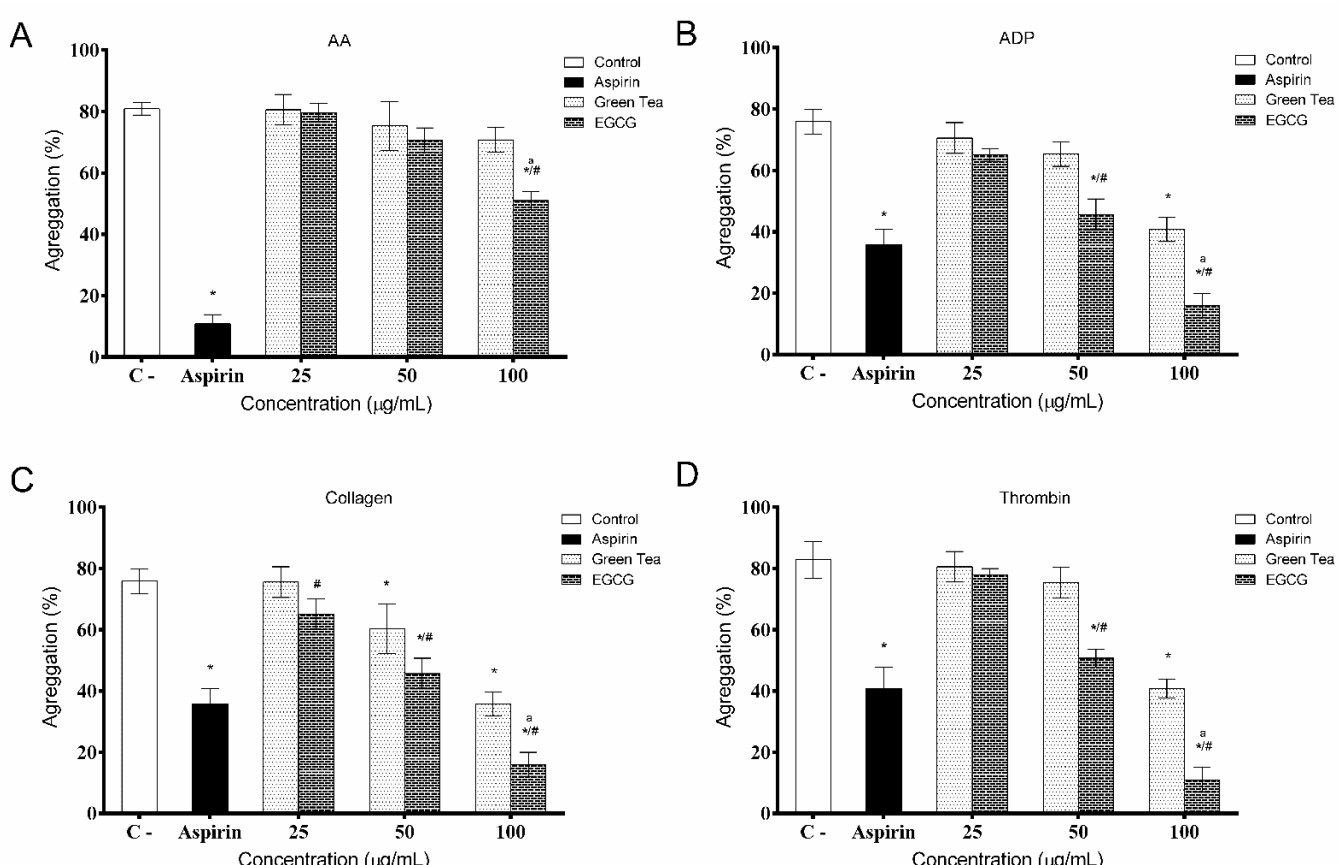

**Figure 1.** Inhibitory effects of EGCG and GTE at concentrations of 25, 50 and 100 μg/mL on platelet aggregation induced by three different agonists (ADP, collagen and thrombin) in platelet-rich plasma (PRP) fraction. (**A**) Aspirin. (**B**) Adenosine diphosphate. (**C**) Collagen. (**D**) Thrombin. Data are expressed as mean ± SD. Comparisons between groups were analyzed by ANOVA and by Tukey's post-test. * $p < 0.05$ compared to negative control group. # $p < 0.05$ compared between different concentrations. [a] $p < 0.05$ compared to positive control group.

Concerning Figure 1B–D, a decrease in platelet aggregation induced by three different agonists (ADP, collagen and thrombin) was observed after treatment with EGCG and GTE. Nonetheless, this inhibitory effect was significant only with EGCG (100 μg/mL). Furthermore, platelets challenged with GTE at 100 μg/mL exhibited an aggregation decrease analogous to the aspirin effect in the presence of ADP, collagen and thrombin. No significant difference regarding the antiplatelet aggregation effect was observed with EGCG and GTE at 25 and 50 μg/mL compared to the positive control, showing $IC_{50}$ = 29.436 ± 1.2 and 25.213 ± 0.9 μg/mL (ADP), $IC_{50}$ = 28.856 ± 1.3 and 24.846 ± 1.1 μg/mL (collagen) and $IC_{50}$ = 27.573 ± 1.0 and 20.032 ± 0.6 μg/mL (thrombin) to GTE and EGCG, respectively.

### 3.2. Secondary Hemostasis (Coagulation)

Table 1 shows data regarding the EGCG and GTE effects on blood coagulation, evaluating the activated partial thromboplastin time (aPTT) and the prothrombin time (PT), corresponding to the coagulation intrinsic and extrinsic pathways, respectively.

Only heparin showed anticoagulant activity at all concentrations assessed. No evidence of the GTE and EGCG action on secondary hemostasis was observed since aPTT and PT values did not differ from the control. Nevertheless, heparin, as a standard positive control, was more efficient in inhibiting clot formation.

**Table 1.** GTE and EGCG effects on blood clotting associated with secondary hemostasis, evaluating activated partial thromboplastin time (aPTT) and prothrombin time (PT).

| Sample | aPPT (s) | PT (s) |
|---|---|---|
| Control | 30.2 ± 2.25 | 13.8 ± 0.43 |
| Heparin (25 µg/mL) | 240 ± 0 | 95.7 ± 0.60 |
| Heparin (50 µg/mL) | 240 ± 0 | 120 ± 0 |
| Heparin (100 µg/mL) | 240 ± 0 | 120 ± 0 |
| Green Tea (25 µg/mL) | 34.7 ± 1.30 | 13.5 ± 0.15 |
| Green Tea (50 µg/mL) | 30.5 ± 3.61 | 14.6 ± 0.39 |
| Green Tea (100 µg/mL) | 30.6 ± 0.51 | 14.7 ± 0.32 |
| EGCG (25 µg/mL) | 35.2 ± 1.25 | 14.4 ± 0.23 |
| EGCG (50 µg/mL) | 35.3 ± 0.86 | 14.1 ± 0.35 |
| EGCG (100 µg/mL) | 32.8 ± 1.10 | 13.5 ± 0.30 |

Results are expressed as mean ± SD.

### 3.3. Cytotoxicity Effects on Melanoma Cell Line B16F10

Figure 2 shows the cell viability of the melanoma cell line B16F10, evaluated by MTT and Alamar Blue® methods, after challenging them with EGCG and GTE. The results obtained by both assays showed a marked cytotoxic effect after exposing cells to EGCG at 100 µg/mL ($p < 0.05$), promoting an average of 58% cell infeasibility compared to the control. Concerning the treatment with GTE at 100 µg/mL, the results indicated a lower cytotoxic effect since GTE reduced the proliferation of melanoma cells by approximately 28% compared to the control and EGCG groups. At a concentration of 50 µg/mL, EGCG also exhibited a significant cytotoxic effect on B16F10 cells compared to the control group, showing $IC_{50} = 23 ± 0.8$ and $29 ± 0.9$ µg/mL to EGCG and GTE, respectively.

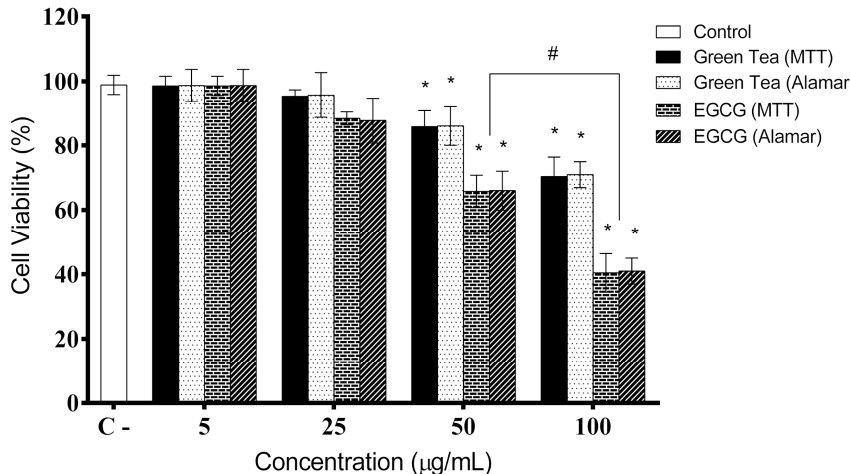

**Figure 2.** Cytotoxicity effects of GTE and EGCG at different concentrations on melanoma cell line B16F10 measured by MTT and Alamar Blue® assays. Cells grown in DMEM medium were used as control group. Data are expressed as mean ± SD. Comparisons between groups were analyzed by ANOVA and by Tukey's post-test. * $p < 0.05$ compared to negative control group. # $p < 0.05$ compared between different concentrations.

### 3.4. Flow Cytometry Apoptosis (Melanoma Cells)

On the basis of cell morphological appearance, no significant apoptotic effect was observed in melanoma B16F10 cells after exposure to different concentrations of GTE and EGCG. The cells exhibited a distribution pattern similar to the control cells grown only in DMEM medium. Nevertheless, the use of cisplatin showed significant apoptotic results concerning the cell size decrease and granularity increase (data not shown). The absence of cell labeling with Annexin or Propidium iodide of cells treated with GTE and

EGCG indicates cellular cytotoxicity not associated with the activation in the early process of apoptosis.

### 3.5. Apoptosis by DAPI Incubation (Melanoma Cells)

Figure 3 depicts the apoptotic effect of GTE and EGCG at a concentration of 100 μg/mL on B16-F10 melanoma cells analyzed by DAPI staining to evaluate nuclear morphological changes. In the control group (Figure 3A,B), the melanoma cells showed a rounded shape and were homogeneously stained. However, after 24 h of exposure to EGCG, the cells showed blebbing nuclei, pyknotic bodies, morphological alterations and granular apoptotic bodies compared to the cell control group. Concerning the GTE effect (Figure 3C,D), despite exhibiting apoptotic characteristics, the nuclear and cellular morphological alterations, such as pyknosis and fragmentation, were less pronounced, whereas the analysis of EGCG-treated cells showed fragmented nuclear structures, condensed cells and reduced sizes (Figure 3E,F).

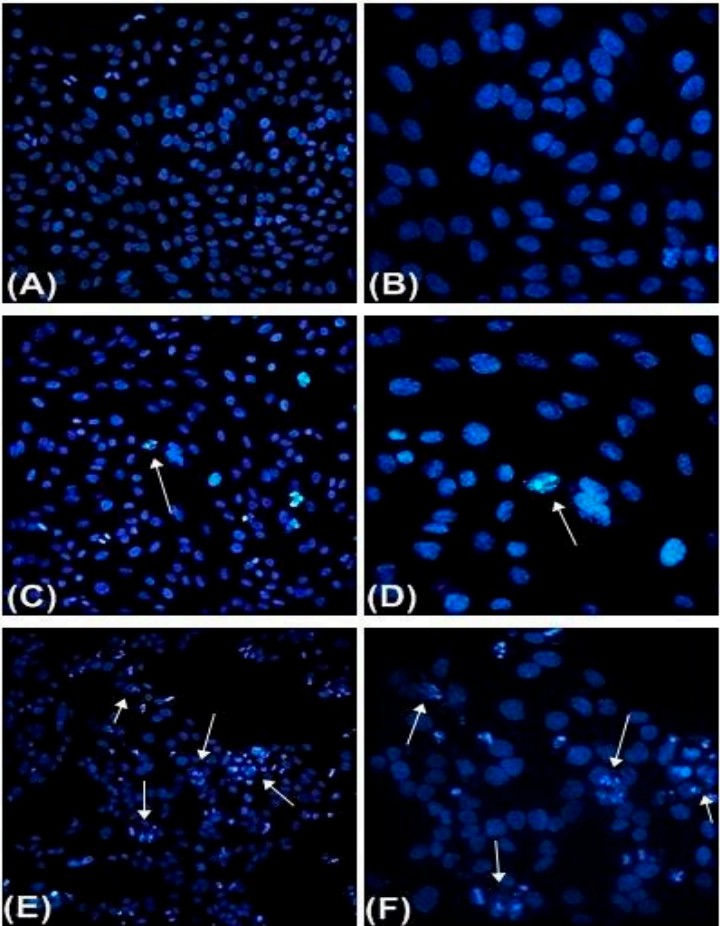

**Figure 3.** Apoptotic effect of GTE and EGCG at 100 μg/mL concentration on melanoma cell line B16F10 after DAPI labeling to assess nuclear morphology. (**A,B**) Untreated B16F10 control cells. (**C,D**) B16F10 cells treated with GTE showing nuclear morphological changes (pyknosis and fragmentation) (arrows). (**E,F**) B16F10 cells treated with EGCG showing marked nuclear morphological changes, such as pyknosis and fragmentation (arrows). Micrographs performed by fluorescence microscopy at magnifications of 20× for (**A,C,E**) and 40× for (**B,D,F**).

These results indicate that the inhibition of proliferation and the decreased viability of melanoma cells after treatment with EGCG and GTE are mainly related to apoptosis. Nonetheless, exposure to EGCG displayed a significant apoptotic effect at the maximal concentration (100 μg/mL) compared to different GTE concentrations.

### 3.6. Melanin Production in Melanoma Cells

The GTE and EGCG effect on melanin production was evaluated by exposing B16-F10 murine melanoma cells at compound concentrations of 25, 50 and 100 µg/mL, as depicted in Figure 4. Both compounds decreased melanin synthesis only in challenged cells at a concentration of 100 µg/mL compared to control cells grown in the DMEM medium. The B16-F10 cell exposure to the maximal EGCG concentration (100 µg/mL) showed an inhibitory effect on melanin synthesis with a mean value of 23.1%, whereas GTE exhibited a mean inhibition of 14.10% at 100 µg/mL, showing $IC_{50}$ = 48.324 $\pm$ 1.2 and 49.432 $\pm$ 1.1 µg/mL to GTE and EGCG, respectively.

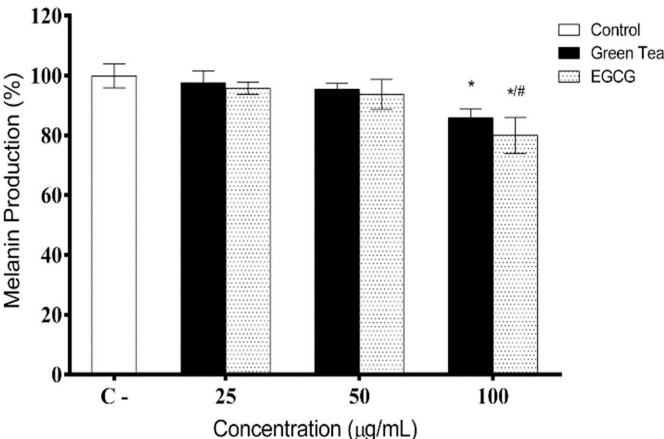

**Figure 4.** Effect of green tea extract and EGCG on melanin production in melanoma cell line B16F10 treated at concentrations of 25, 50 and 100 µg/mL for 48 h. Cells grown in DMEM were used as a negative control. Comparisons between groups were analyzed by ANOVA and by Tukey's post-test. * $p < 0.05$ compared to negative control group. # $p < 0.05$ compared between different concentrations.

### 3.7. Angiogenesis (Endothelial Cells)

Concerning the cell viability of endothelial cells (RAEC), no cytotoxicity was evidenced after cell exposure to GTE and EGCG at the different concentrations tested (25, 50 and 100 µg/mL) (Figure 5). Showing $IC_{50}$ = > 100 µg/mL to GTE and EGCG, respectively.

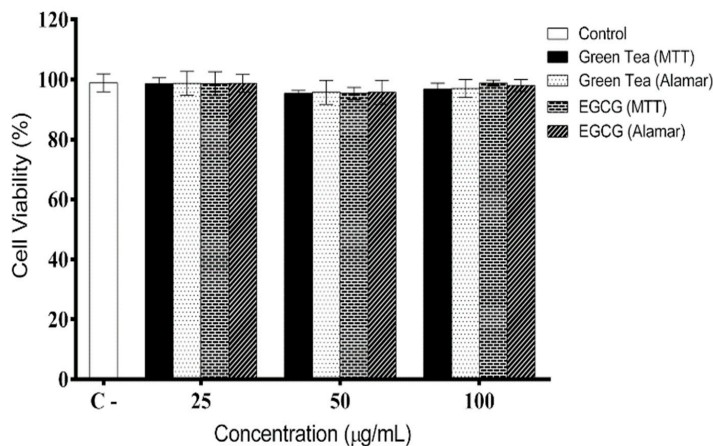

**Figure 5.** Effect of GTE and EGCG at concentrations of 25, 50 and 100 µg/mL on endothelial cell (RAEC) viability determined by MTT and Alamar Blue® assays. Comparisons between groups were analyzed by ANOVA and by Tukey's post-test.

Respecting the angiogenic capacity, Figure 6 A,B shows the ability of GTE and EGCG to significantly inhibit the capillary formation from endothelial cells in a concentration-

dependent manner ($p < 0.001$) compared to the control (PBS) It is noteworthy that the antiangiogenic effect induced by EGCG at 100 µg/mL was higher than that exhibited by GTE, although significant effects were also observed at 25 and 50 µg/mL on the tubular structures count (Figure 6B). Nonetheless, both compounds exhibited the ability to reduce the formation of new blood vessels. Showing $IC_{50} = 25.54 \pm 1.0$ and $14.47 \pm 0.6$ µg/mL to GTE and EGCG, respectively.

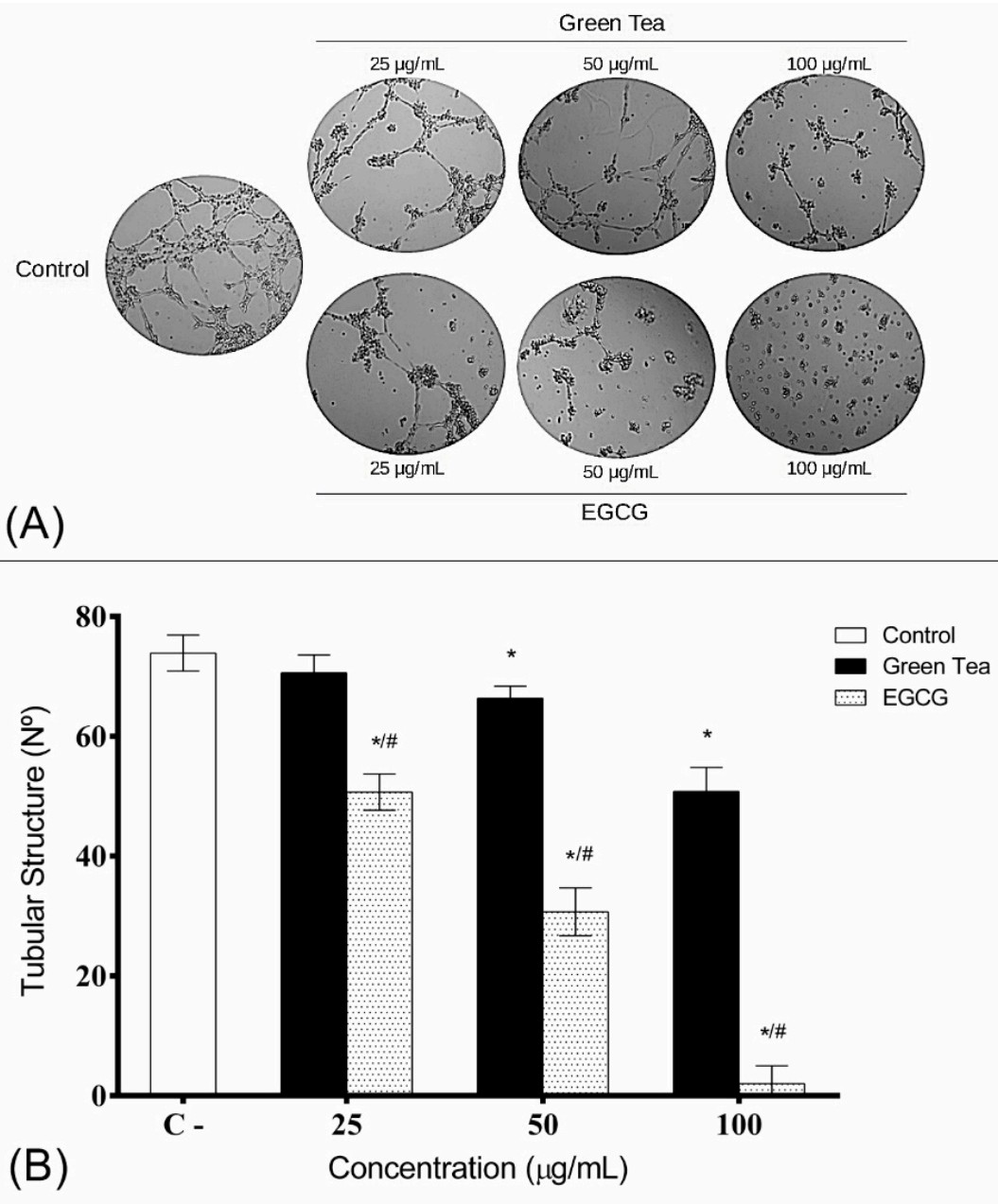

**Figure 6.** Angiogenic potential effect of GTE and EGCG at 25, 50 and 100 µg/mL to evaluate the formation of capillary structure in Matrigel, examined under inverted light microscopy (100×) (**A**). The number of capillary structures was determined using software for image analysis (**B**). Culture medium F12 was used as a positive control. *p*-values represent the results of statistical analysis (One-way ANOVA). * $p < 0.05$ compared to negative control group. # $p < 0.05$ compared between different concentrations.

### 3.8. Inhibition of Vascular Endothelial Growth Fator (VEGF) Secretion (Endothelial Cells)

The evaluation of the effect of GTE and EGCG in inhibiting VEGF secretion in healthy endothelial cells and cells stimulated with TNFα is shown in Figure 7. The results indicate that cellular exposure to these compounds promoted a significant decrease in VEGF secretion with both GTE and EGCG compared to controls. EGCG at concentrations of 50 and 100 μg/mL was efficient in reducing VEGF secretion by around 50% and 60%, respectively, compared to the control. Regarding the GTE effect, the concentrations of 50 and 100 μg/mL showed a reduction of ~40% in the VEGF secretion inhibition, revealing satisfactory values respecting both negative controls, mainly the TNFα stimulated control. Showing $IC_{50}$ = 28.654 ± 1.1 and 22.768 ± 0.9 μg/mL to GTE and EGCG, respectively.

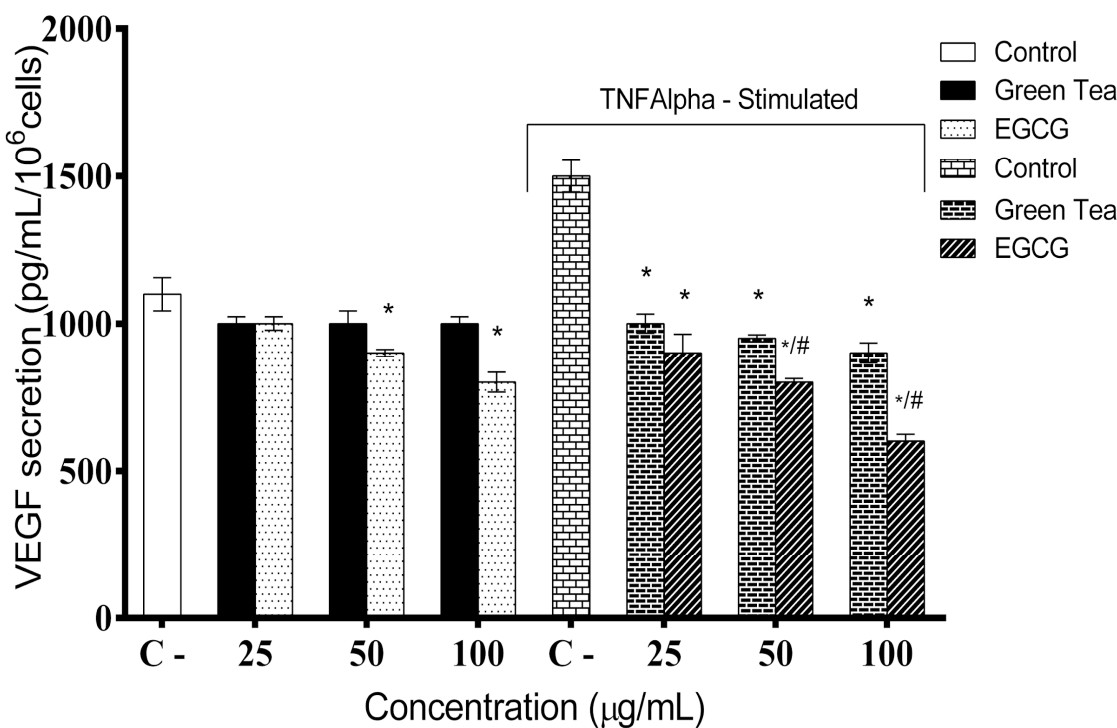

**Figure 7.** Effect of vascular endothelial growth factor (VEGF) inhibition on RAEC cells individually exposed to GTE and EGCG at concentrations of 25, 50 and 100 μg/mL. Comparisons between groups were analyzed by ANOVA and by Tukey's post-test. * $p < 0.05$ compared to negative control group. # $p < 0.05$ compared between different concentrations.

The inhibiting result of VEGF secretion from green tea and EGCG showed a positive effect, mainly with EGCG (100 ug), although no significant inhibition value was observed with GTE compared to EGCG, the major compound present in tea.

### 3.9. Inhibition of IL-8 Secretion (Endothelial Cells)

The evaluation of GTE and EGCG effects at 25, 50 and 100 μg/mL on IL-8 secretion inhibition in healthy endothelial cells and TNFα-stimulated cells is depicted in Figure 8. Overall, the inhibitory action of these compounds showed a concentration-dependent reduction of this secretion. The results showed that EGCG at 50 and 100 μg/mL significantly inhibited IL-8 secretion by approximately 75% and 46%, respectively while GTE promoted ~25% of inhibition of IL-8 secretion at 100 μg compared to the control. Showing $IC_{50}$ = 32.076 ± 1.5 and 21.654 ± 1.2 μg/mL to GTE and EGCG, respectively.

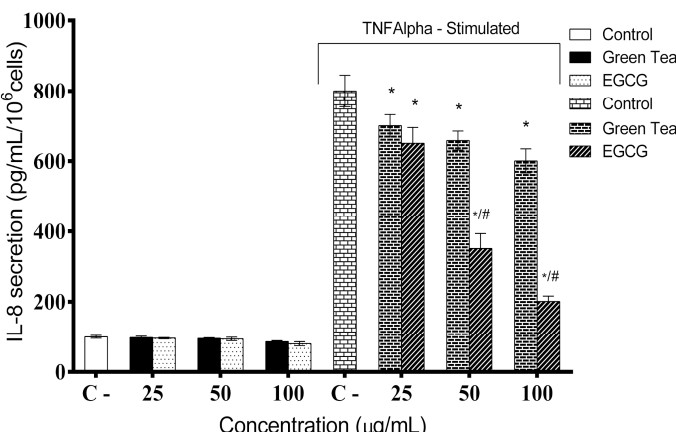

**Figure 8.** Inhibition of IL-8 secretion on RAEC cells individually exposed to GTE and EGCG at concentrations of 25, 50 and 100 μg/mL. Comparisons between groups were analyzed by ANOVA and by Tukey's post-test. * $p < 0.05$ compared to negative control group. # $p < 0.05$ compared between different concentrations.

## 4. Discussion

The melanoma incidence and its pathogenesis signaling represent a cornerstone in therapeutic approaches research since conventional therapeutic options lead to several side effects in patients, such as a decrease in immune system function, therapy resistance and thromboembolism [13,16]. On the basis of these aspects, therapies involving medicinal plant use have shown a tendency to prospect effective anticancer compounds with low side effects for clinical treatment [8,18]. Thence, the chemical diversity of medicinal plants constitutes a bioactive compound source, whose potential application as anticoagulant and anticancer agents stimulates several studies [21,24].

In this regard, green tea possesses several bioactive compounds closely related to human health, such as catechins, flavonoids, anthocyanins and phenolic compounds. Studies have reported several health benefits, such as anticancer, antithrombotic and immunological protective effects [29,30,42]. Despite the literature showing green tea and EGCG properties, few studies have analyzed their potential application in malignant melanoma treatment. Thereby, this study reports the beneficial effect of GTE and EGCG in the treatment and minimization of side effects described in conventional therapy.

Overall, the in vitro biological effects observed after challenge with GTE and EGCG, such as platelet aggregation inhibition, blood clotting and tubular structure reduction, may be due to the synergistic action between various polyphenolic components. This cooperation may have improved the performance of these compounds, promoting these achieved promising results [43]. Indeed, the synergy is related to a secondary metabolite mixture, mainly polyphenols, present in GTE, whose interaction can result in a robust action applied in association with drugs to enhance inhibitory effects in several processes. Indeed, the synergy can be related to a secondary metabolite mixture, mainly polyphenols, present in GTE, whose interaction can result in a robust form to be applied in association with drugs to enhance inhibitory effects in several pathological processes and help in their treatments [44,45].

The thrombosis risk is one of the adverse effects of conventional oncological therapies, which reinforces the plant secondary metabolite prospection as an approach for clinical use. Thereby, both EGCG and GTE exhibited a significant antiplatelet effect in assessing the hemostatic mechanism. Although the aspirin antiplatelet action was superior, EGCG, the main green tea polyphenol, was efficient in influencing the primary hemostasis stage by decreasing platelet aggregation induced by agonists. Similar results with EGCG were reported by Joo et al. [35] and Sinegre et al. [46], showing a significant inhibition of platelet aggregation associated with a decrease in agonist agents (ADP and collagen). Furthermore, according to Sinegre et al. [46], EGCG attenuated the formation of ROS and TXB2,

also inhibiting NADPH and significantly suppressing agonists, which may explain the consequent decrease in platelet aggregation.

This effect can be explained based on the irreversible acetylation of cyclooxygenase promoted by EGCG, which prevents the formation of thromboxane A2 from the arachidonic acid presence, which can decrease the required change in glycoprotein IIbIIIa for platelet aggregation [35,46].

Regarding coagulation, both EGCG and GTE results suggest the compound-specific action on factors and processes involved in the intrinsic pathway of blood clotting since a significant inhibitory activity on coagulation was evidenced by aTPP and PT pathway assays (secondary hemostasis) [47]. Despite the secondary metabolite-rich content (flavonoids and catechins) of green tea, its inhibitory effect on coagulation in the extrinsic pathway was unsatisfactory compared to EGCG. Misztal et al. [48] described a similar result using EGCG at concentrations >10 μM, which was able to normalize coagulation against the inhibition caused by hypochlorous acid generated by neutrophils in the fight against injury. Nevertheless, these authors established no correlation between the results and the coagulation pathways.

The EGCG and GTE results are relevant, although heparin exhibited greater anticoagulant efficiency, triggering intrinsic pathway factors in treatments associated with thromboembolism, a side effect of cancer treatment [49]. Despite its effectiveness in maintaining hemostasis, heparin causes adverse effects in clinical therapy, such as bleeding, thrombocytopenia and hypersensitivity [50]. This encourages the anticoagulant agent prospection in medicinal plants [20,25], whose studies are still scarce.

Besides the antiplatelet effect, cell viability by MTT and Alamar Blue® tests evidenced the GTE and EGCG cytotoxic capacity on B16F10 melanoma cells. Under experimental conditions, data displayed that these compounds exerted an antiproliferative effect since the viability decrease indicates a significant cell growth inhibition, mainly after exposure to EGCG compared to GTE, both at 100 μg/mL. Overall, studies have reported the analogous antiproliferative effects of EGCG in colorectal cancer cells [51], the decrease and morphological alteration of melanocytes [52] and also apoptosis induction on lung and breast cancer cells [53,54]. Regarding green tea, antiproliferative activity has been described against different types of cancers due to its chemical composition [29,30].

Cytotoxicity indicated by a significantly decreased growth of B16F10 melanoma cells exposed to different concentrations of EGCG and GTE showed no significant apoptotic effect. The similar pattern between challenged EGCG and GTE cells and control cells indicates the absence of Annexin-V/FITC/Propidium iodide labeling. Thence, cellular cytotoxicity determined by MTT and Alamar Blue® in cells treated with EGCG and GTE is not associated with an early phase of apoptosis.

Nevertheless, cell staining by DAPI after treatment with these compounds showed specific morphological and structural changes, including fragmented nuclear structure and decreased cell size. Blebbing nuclei, pyknotic bodies and morphological alterations are indicative of apoptotic induction [55]. The marked apoptotic effect at maximal EGCG concentration compared to green tea and the control suggests cell death caused by late apoptosis. A similar result with DAPI staining of BGC-823 gastric cancer cells treated with EGCG was reported by Xue et al. [56], further indicating the combination of EGCG with other drugs to increase the treatment sensitivity to incapacitate carcinogenic cells.

Melanogenesis is another parameter to be evaluated at the cellular level concerning a compound with a potential anti-melanoma effect since melanin plays a key role in influencing melanoma development [10]. Thus, GTE and, specifically, EGCG exhibited a concentration-dependent inhibitory effect on melanin production compared to non-treated cells. Despite the low inhibitory effect, the concentration-dependent interference of these compounds in the melanin content indicates the possibility that higher concentrations could further decrease this synthesis in skin tumor cells. Similar results have been described in several studies, showing EGCG's ability to reduce melanin production in melanoma cells by approximately 30% [57–59]. Therefore, EGCG and other flavonoids display a regulatory

capacity on melanin secretion and, consequently, potential for the prevention and treatment of melanoma [60,61].

Concerning the green tea extract and EGCG effects on endothelial cells, no cytotoxicity was observed through cell viability assays. Nevertheless, cell exposure to GTE and EGCG showed the concentration-dependent antiangiogenic potential through the significant reduction in tubular structures, mainly with EGCG at 100 µg/mL. The inhibition of new blood vessel formation is important because sustained angiogenesis is one of the oncogenic hallmarks that plays a crucial role in the initiation, maintenance and progression of cancer since the endothelial cell organization allows for rapid tumor growth [62].

Considering that neovascularization is closely associated with tumor angiogenesis, the inhibition of vascular endothelial growth factor (VEGF) is essential to prevent endothelial cell proliferation [63,64]. Therefore, the significant inhibition of VEGF secretion in TNF$\alpha$-stimulated endothelial cells indicates the antiangiogenic effect of both GTE and EGCG, although EGCG displayed a superior inhibitory performance on VEGF secretion. Several studies with cancer cells have reported the inhibition of VEGF secretion by green tea and EGCG by decreasing the expression of both mRNA and protein levels [65–67], highlighting green tea in the prevention of tumor angiogenesis and metastasis [68].

Furthermore, GTE and EGCG exhibited a significant inhibitory effect on IL-8 cytokine secretion, although cell exposure to EGCG showed more efficient results, evidencing its protective role in reducing inflammatory responses. This cytokine inhibition may also be closely linked to VEGF secretion, antiangiogenic effect and cellular morphological changes observed over several treatments since IL-8 participates in the maintenance of balance between physiological reactions and pathological bodily processes. IL-8 expression is associated with tumors, and its accumulation has been detected in infiltrating neutrophils, macrophages and cancer and endothelial cells [69]. Therefore, IL-8 may be a significant regulatory factor in the tumor microenvironment, which should be explored as a target in pathogenesis, prognosis and therapy due to its accumulation in cancers [70]. Overall, data support the beneficial effects of green tea's bioactive compounds, whose promoting effects support its use as an approach to controlling physiological and metabolic problems [71,72].

## 5. Conclusions

In summary, this study showed the in vitro antithrombotic, antitumor and antiangiogenic properties of green tea extract and epigallocatechin-3-gallate. The data evidenced that both GTE and EGCG displayed a significant inhibition of platelet aggregation, acting on coagulation through primary homeostasis. Moreover, GTE and, mainly, EGCG exhibited antitumor effects on melanoma cells evaluated through an antiproliferative effect due to late apoptosis, as well as a concentration-dependent inhibitory effect on melanin production. Furthermore, regarding angiogenesis, the results showed antiangiogenic potential through a significant reduction in tubular structures. Both compounds were able to promote a marked inhibition in the production of VEGF and IL-8, mainly in cells stimulated by TNF-$\alpha$. Altogether, the results with EGT and EGCG suggest the possibility of developing formulations to assist in antitumor therapy, aiming to attenuate the adverse reactions of conventional treatments. Although further studies are required, data provide promising evidence supporting the potential application of EGT and EGCG as an alternative to assist oncologic treatments.

**Author Contributions:** Conceptualization, J.R.D.d.L., M.P.F. and M.d.G.A.; methodology, J.R.D.d.L. and G.A.-S.; formal analysis, J.R.D.d.L. and S.V.e.S.; investigation, J.R.D.d.L., S.V.e.S., R.M.d.S. and M.P.F.; resources, J.R.D.d.L., M.P.F., M.d.G.A., J.A.L. and G.A.-S.; data curation, J.R.D.d.L., G.A.-S., J.A.L. and M.d.G.A.; writing—original draft preparation, R.M.d.S., M.P.F., J.R.D.d.L., G.A.-S., J.A.L. and M.d.G.A.; writing—review and editing, J.A.L., J.R.D.d.L., G.A.-S. and M.d.G.A.; supervision, J.R.D.d.L., G.A.-S., J.A.L. and M.d.G.A.; project administration, G.A.-S.; funding acquisition, M.d.G.A. and G.A.-S. All authors have read and agreed to the published version of the manuscript.

**Funding:** This research was funded by JBS Fund for the Amazon (Ordinance No. 348/2021-UEAP), Research Support Foundation of the State of Amapá-FAPEAP (grant No. FAPEAP/Decit/SCTIE/MS/ SESA-AP/CNPq No. 003/2020) and the Federal University of Rio Grande do Norte (grant No. 397/2020).

**Institutional Review Board Statement:** Not applicable.

**Informed Consent Statement:** Not applicable.

**Data Availability Statement:** Not available.

**Acknowledgments:** The authors would like to thank the CNPq for providing a postgraduation fellowship (process no. 169246/2018-3), the Coordination for the Improvement of Higher Education Personnel (CAPES) (Finance Code 001) and the Research Center for Redox Processes in Biomedicine at the University of São Paulo (USP) for their technical assistance in cell cultures.

**Conflicts of Interest:** The authors declare no conflict of interest.

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
