# Peer review of "In Vitro Antithrombotic, Antitumor and Antiangiogenic Activities of Green Tea Polyphenols and Its Main Constituent Epigallocatechin-3-gallate"

_processes, doi:10.3390/pr11010076_

Round 1
Reviewer 1 Report
The article on “In Vitro Antithrombotic, Antitumor and Anti-angiogenic Activities of Green Tea Polyphenols and its Main Constituent Epigallocatechin-3-gallate” is quite interesting research work proved that polyphenols from green tea and Epigallocatechin-3-gallate has diverse biological activity.
However, there few questions authors need to be address
1. Is the authors analysed and characterized the polyphenols from green tea or based on literature simply done the experiment with green tea extract.
2. Authors advised to include the thrombin inhibition activity of green tea extract and Epigallocatechin-3-gallate
3. No. of tubular structures should be calculated for capillary formation in RAEC cells.
4.Include the catalog.no for green extract polyphenols and Epigallocatechin-3-gallate from sigma
Reviewer 2 Report
The manuscript submitted for review describes the properties of green tea and EGCG in terms of biological activity, such as anti-cancer, antiangiogenic, etc. In my opinion, the manuscript was well designed and the methods used to assess activity were well chosen. In my opinion, however, a rather important issue has been overlooked and should be discussed.
There is no standardization of the green tea extract in terms of the presence of ECGC, which makes the results difficult to compare. How do the authors know whether the effect of the tea is not due to the ECGC content in it or maybe it is the effect of a synergistic effect resulting from the presence of other polyphenols.
I believe that this point is very important and should be justified in the manuscript.
minor remarks:
· has the green tea extract been previously dried (freeze-dried)?
· statistical elaboration is quite poor, the authors have the option of using more sophisticated ones, e.g. the IC50 value
· not all protocols specify which solvent was used to dissolve the GTE and EGCG
Round 2
Reviewer 2 Report
All my remarks have been addressed satisfactory.
The manuscript has been significantly impoved and I recomend to publish it in present form.
Congratulations to the Autors